# The Function of the Kynurenine Pathway in the Placenta: A Novel Pharmacotherapeutic Target?

**DOI:** 10.3390/ijerph182111545

**Published:** 2021-11-03

**Authors:** Michelle Broekhuizen, A. H. Jan Danser, Irwin K. M. Reiss, Daphne Merkus

**Affiliations:** 1Division of Pharmacology and Vascular Medicine, Department of Internal Medicine, Erasmus University Medical Center, 3015 GD Rotterdam, The Netherlands; a.danser@erasmusmc.nl; 2Division of Neonatology, Department of Pediatrics, Erasmus University Medical Center, 3015 GD Rotterdam, The Netherlands; i.reiss@erasmusmc.nl; 3Division of Experimental Cardiology, Department of Cardiology, Erasmus University Medical Center, 3015 GD Rotterdam, The Netherlands; d.merkus@erasmusmc.nl; 4Walter Brendel Center of Experimental Medicine, University Clinic Munich, LMU Munich, 81377 Munich, Germany

**Keywords:** tryptophan, kynurenine, indoleamine 2,3-dioxygenase, placenta, pregnancy, therapy

## Abstract

(*L*-)tryptophan is metabolized via the kynurenine pathway into several kynurenine metabolites with distinct functions. Dysfunction of the kynurenine pathway can lead to impairments in vascular regulation, immune regulation, and tolerance. The first and rate limiting enzyme of this pathway, indoleamine 2,3-dioxygenase (IDO), is highly expressed in the placenta and reduced in placentas from complicated pregnancies. IDO is essential during pregnancy, as IDO inhibition in pregnant mice resulted in fetal loss. However, the exact function of placental IDO, as well as its exact placental localization, remain controversial. This review identified that two isoforms of IDO; IDO1 and IDO2, are differently expressed between placental cells, suggesting spatial segregation. Furthermore, this review summarizes how the placental kynurenine pathway is altered in pregnancy complications, including recurrent miscarriage, preterm birth, preeclampsia, and fetal growth restriction. Importantly, we describe that these alterations do not affect maternally circulating metabolite concentrations, suggesting that the kynurenine pathway functions as a local signaling pathway. In the placenta, it is an important source of de novo placental NAD^+^ synthesis and regulates fetal tryptophan and kynurenine metabolite supply. Therefore, kynurenine pathway interventions might provide opportunities to treat pregnancy complications, and this review discusses how such treatment could affect placental function and pregnancy development.

## 1. Introduction

Pregnancy is a unique condition that allows an allogeneic fetus to grow inside a mother without eliciting an immune response. The major facilitator of the tolerogenic environment is the placenta, where the mother and fetal tissue are in direct contact to allow the transport of oxygen and essential nutrients from mother to fetus and the removal of CO_2_ and waste products from the fetal circulation. Development of the placenta starts just after blastocyst implantation, already before the embryo is formed. The placenta grows during gestation, and in its mature form, it consists of large fetal vascular networks inside villous trees that are lined by the fetal cytotrophoblasts and syncytiotrophoblasts (Figure 1). The latter cells form the direct interface between the maternal and fetal circulation as the fetal villi bath in the placental intervillous space that is filled with maternal blood. Due to this structure, nutrients and oxygen should pass the syncytiotrophoblasts, and fetal endothelial cells to reach the fetal circulation.

Tryptophan (*L*-Tryptophan) is an essential amino acid, of which a small amount is used for protein synthesis, however, most tryptophan is metabolized through the kynurenine pathway. The kynurenine pathway is highly conserved between species, underlining its evolutionary importance. It generates more than 10 different metabolites with distinctive functions, including modulation of the immune system and regulation of vascular function [1]. A global overview of the kynurenine pathway is shown in Figure 2. The rate limiting step in the kynurenine pathway is the conversion of tryptophan to kynurenine (*L*-kynurenine), which is catalyzed in the liver by tryptophan 2,3-dioxygense (TDO) and extrahepatically by indoleamine 2,3-dioxygenase (IDO). It was only in 2007 that IDO was discovered to exist in two different isoforms, which were named IDO1 and IDO2 [2,3]. Hence, in studies prior to 2007, these two isoforms were not distinguished. In the remainder of this review, we will refer to the specific isoform when possible, and IDO will be used if no distinction was made between isoforms.

The first report on the potential pathological role of the kynurenine pathway dates back to 1956, when Boyland and Williams reported increased concentrations of multiple kynurenine pathway metabolites in the urine of patients with cancer of the bladder [4]. This initial discovery has led to the development of inhibitors of IDO [5]. Only more recently has the importance of the kynurenine pathway and IDO emerged in other fields, including pregnancy and placental development. The importance of IDO1 during pregnancy was discovered in the 1990s when treatment of pregnant mice with the IDO inhibitor 1-methyl-tryptophan from the start of pregnancy resulted in fetal loss [6].

Whereas IDO1 is only upregulated through interferon (IFN)-γ in most tissues [7,8], it is constitutively expressed under physiological conditions in the placenta [9,10]. Munn et al. [6] initially proposed that placental IDO activity prevents immune activation, and creates a tolerogenic environment by preventing T cell activation through the depletion of tryptophan [6,11]. However, it is now generally believed that IDO acts through the formation of active kynurenine metabolites and oxidized nicotinamide adenine dinucleotide (NAD^+^) which directly affect immune cells, rather than locally deplete tryptophan [12,13]. The molecular mechanisms involved in IDO function have recently been reviewed by Yeung et al. [14]. Most research has focused on immune regulation, but the kynurenine pathway is also involved in other processes, including neovascularization [15,16], vasodilation [17,18,19], anti-oxidative processes [20,21,22], and the regulation of apoptosis and endothelial dysfunction [23]. These processes are all important determinants of placental development and are consequently required for a healthy pregnancy. Indeed, other reviews have suggested a role for disturbances in the kynurenine pathway in placental pathologies [22,24,25]. Research has shown that tryptophan metabolism can modify the behavior of T cells and alter vascular tone in vitro, but the actual relevance of the placental kynurenine pathway in vivo remains speculative. The presence of specific kynurenine pathway enzymes determines which metabolites are formed in tissue [9], and may thus help to provide clues with regard to the functional roles of the placental kynurenine pathway. The regulation of the kynurenine pathway and its enzymes has been summarized in a review by Badawy [9]. The following sections will describe the current knowledge on the cell type specific expression of kynurenine pathway enzymes and transporters in the placenta using previously published immunohistochemistry and single cell RNA sequencing data. This review also covers how kynurenine pathway alterations might contribute to pregnancy-related pathologies, and how this could be targeted therapeutically in the future.

## 2. Placental Expression of Kynurenine Pathway Enzymes

IDO1 is a cytoplasmic enzyme which occurs at high levels in the placenta compared to other organs [10,26]. The human placenta expresses both IDO1 and IDO2, and both are increased at term compared to the first trimester placenta [19,27]. In accordance with expression data, IDO activity increases from week 14 until term in the human placenta [27,28].

The kynurenine/tryptophan ratio in plasma has commonly been used as a measure for placental IDO1 activity under the assumption that circulating kynurenine and tryptophan concentrations depend solely on placental synthesis and release. However, using the ex vivo human cotyledon perfusion model we have shown that the addition of tryptophan to the maternal circulation did not affect the release of kynurenine metabolites into either the maternal and fetal circulation over a period of three hours [19]. Moreover, kynurenine metabolites were found to be approximately equal in the afferent human umbilical artery and efferent umbilical vein in vivo (Table 1). Given these data, it seems that the maternal kynurenine metabolite levels do not represent placental IDO activity. Interestingly, all kynurenine metabolites except anthranilic acid were higher in the fetal compared to the maternal circulation (Table 1). Since these do not result from placental IDO1 activity, these metabolites should be actively transported from the maternal to the fetal circulation, and/or formed in the fetus itself. Since placental function is unlikely to influence circulating metabolites, in the remainder of this review we will focus on placental kynurenine pathway functions, and not on alterations in circulating metabolites.

Since the discovery of IDO, several immunohistochemical studies have investigated the expression of IDO in the placenta, unfortunately with conflicting results (Table 2). Most studies have used an antibody from the lab of Dr. Takikawa, Kawasaki Medical School, Okayama Japan, which was produced by injecting purified human placenta derived IDO into BALB/c mice [38]. With this antibody, no distinction can be made between IDO isoforms. Recently, Kudo et al. [39] provided a clear separation between IDO1 and IDO2 expression using specific antibodies, and Murthy et al. [40] showed increasing expression of IDO1 from the second trimester to term in endothelial cells only. Although several studies were performed using the antibody provided by Takikawa, the immunohistochemical picture varied markedly between studies. The findings by Santoso et al. [41], Sedlmayr et al. [42], and Blaschitz et al. [43] seem to resemble IDO1 expression, whereas the findings by Kamimura et al. [28], Kudo et al. [44] and Ligam et al. [45] agree with IDO2 expression Particularly based on the most recent research by Kudo et al. [39] and Murthy et al. [40], we conclude that in the first trimester, IDO1 is expressed in glandular epithelial cells of the decidua, while at term, it is mainly expressed in decidual macrophages and placental endothelial cells. IDO2 is continuously expressed in syncytiotrophoblasts from the first trimester until term, with some additional expression in extravillous trophoblasts.

Apart from IDO1 and IDO2, the placenta expresses the tryptophan catabolizing enzyme TDO, albeit its expression is very low compared to IDO1 and is therefore unlikely to play an important role in placental kynurenine pathway function [19,27]. The placenta also expresses most of the other kynurenine pathway enzymes noted in Figure 2 [19,27,48]. To further investigate the cell-specific expression of the kynurenine pathway enzymes we scrutinized previously published single cell RNA sequencing (scRNAseq) databases from human placentas [49,50,51]. Vento-Tormo et al. [49] investigated the feto-maternal interface during the first trimester, while Pique-Regi et al. [50] and Tsang et al. [51] investigated term placentas, allowing the identification of the placental kynurenine pathway throughout gestation. We received the complete processed dataset from Pique-Regi et al. [50]. For the original data sets of Tsang et al. [51] and Vento-Tormo et al. [49], we performed normalization, scaling and clustering ourselves. The published metadata were used to trace back the original cell cluster definitions.

Surprisingly, most immune cells; T cells, B cells, natural killer (NK) cells, macrophages (including Hofbauer cells), did not express any of the kynurenine pathway enzymes. Moreover, all three studies showed that there is not a single cell type in the placenta that expresses all enzymes of the kynurenine pathway (Figure 3), suggesting that metabolites must be transferred between neighboring cell types or derived from the systemic circulation. The scRNAseq data of Vento-Tormo et al. [49] showed that IDO1 is mainly expressed in epithelial cells in the first trimester. At term, IDO1 is mainly expressed by placental endothelial cells as depicted by the database of Tsang et al. [51] (Figure 3B). No vascular endothelial cell population was identified in the database of Pique-Regi et al. [50] (Figure 3C). However, this database revealed some additional IDO1 expression in hematopoietic stem cells (HSC) and endometrial cells. A closer look into the HSC population in this study revealed that endothelial markers von Willebrand factor, VE-cadherin, and CD34 were highly expressed in this population as well (40–60% of the cells), suggesting that HSC population included endothelial cells within this study. IDO2 and TDO were hardly expressed in the placenta in the first trimester, nor at term.

Kynurenine 3-monooxygenase (KMO), kynureninase (KYNU) and 3-hydroxyanthranilate 3,4-dioxygenase (HAAO) were expressed by a small portion of syncytiotrophoblasts both in first trimester (Figure 3A) and in term placentas (Figure 3B,C). The enzyme quinolinate phosphoribosyltransferase (QPRT) was the highest expressed enzyme of the kynurenine pathway, and specifically present in villous cytotrophoblast, and extravillous cytotrophoblast in the first trimester placenta (Figure 3A) and in the extravillous trophoblast and cytotrophoblasts at term (Figure 3B,C). QPRT uses quinolinic acid as a substrate to form NAD^+^ and is therefore essential for de novo NAD^+^ synthesis. The high QPRT expression suggests that cytotrophoblast are major NAD^+^ consumers. Although initially syncytiotrophoblasts were expected to be metabolically most active due to their barrier function, lack of QPRT in these cells, as well as high QPRT expression and de novo NAD^+^ synthesis in cytotrophoblasts are in agreement with a study that showed cytotrophoblasts to have the highest metabolic activity [52]. Both aminoadipate aminotransferase (KAT-2) and kynurenine aminotransferase 3 (KAT-3) can form kynurenic acid and xanthurenic acid, but placental mRNA expression of KAT-2 is relatively low as compared to KAT-3 as measured by scRNAseq (Figure 3) and quantitative reverse transcription polymerase chain reaction [19]. An overview of the expression of kynurenine pathway enzymes is summarized in Figure 4.

To further explore kynurenine pathway enzyme activity, metabolite levels were measured in placental tissue as well as placental explants exposed to tryptophan. Placental explants were shown to take up tryptophan from the medium and secrete significant concentrations of kynurenine, 3-hydroxyanthranilic acid, quinolinic acid and picolinic acid back into the medium, but the kynurenic acid concentration was very low [45,48]. Similarly, relatively low kynurenic acid, anthranilic acid and serotonin concentrations were measured in placental tissue and xanthurenic acid was undetectable [19]. This suggests that these metabolites do not have prominent functions in the placental kynurenine pathway. It should be noted, however, that in ex vivo placenta perfusion experiments, even in the absence of placental tissue, tryptophan spontaneously decomposed into kynurenine, kynurenic acid and anthranilic acid. Therefore, future studies investigating the production of kynurenine metabolites in vitro requires the inclusion of a time-matched control without cells or tissue, to control for spontaneous decomposition of tryptophan and kynurenine pathway metabolites.

Taken together, gene expression, immunohistochemical and metabolite data suggest that, in the placenta, tryptophan is mainly used as a source for de novo NAD^+^ synthesis, through subsequent formation of kynurenine, 3-hydroxykynurenine, 3-hydroxyanthranilic acid, quinolinic acid and NAD^+^.

## 3. Placental Tryptophan Transport

Since IDO1 is a cytosolic enzyme, tryptophan needs to enter the cell to be metabolized. During pregnancy, the plasma levels of tryptophan decrease substantially. It is important to note that plasma levels of tryptophan comprise both free tryptophan as well as tryptophan bound to albumin. The decrease of total tryptophan during pregnancy is accompanied by an increase of the free fraction of tryptophan, due to a decreased albumin concentration [53], and an increased concentration of non-esterified fatty acids (NEFA), a physiological displacer of albumin-bound tryptophan [54,55]. Only the free fraction of tryptophan is available for transport across the placenta to the fetal circulation. Studies that compared tryptophan concentrations between the maternal circulation and umbilical cord blood showed that total tryptophan [30,31,32,33,34,36,56], as well as free tryptophan [35,37] were higher in the fetal compared to the maternal circulation, implying active transport over the placenta. Potential transporters are the large neutral amino acid transporter 1 (SLC7A5/LAT1), the large neutral amino acid transporter 2 (SLC7A8/LAT2), and the sodium-dependent neutral amino acid transporter (SLC1A5/ASCT2). It should be noted that mRNA expression of LAT1, LAT2, and ASCT2 appears to be low in term placentas based on the scRNAseq studies (Figure 3). Nevertheless, LAT1, LAT2 and ASCT2 proteins were all detected with distinct spatial patterns using immunohistochemistry [57], suggesting that new synthesis of these transporters is limited, and that turnover is low as they remain persistently present as a membrane bound transporter. Expression of Slc7a5 and Slc7a8 increased throughout gestation in mice, with a concomitant increase in placental tryptophan content [58]. In human chorionic plate arteries, LAT1 expression is higher compared with LAT2 and ASCT2 [19]. Moreover, LAT1 was shown to facilitate tryptophan uptake in fibroblasts and in the placenta [59,60,61]. The expression of known kynurenine pathway enzymes, transporters and the aryl hydrocarbon receptor (AHR) is summarized in Figure 4. However, placental tryptophan uptake possibly involves other, yet unidentified high-affinity specific transporter systems as well [60,62,63]. Identification of these transporters is highly relevant since tryptophan transport, in addition to IDO1 expression, may also be limiting for its metabolism through the kynurenine pathway [19,64].

The segmented expression of kynurenine pathway enzymes as identified in the previous section would require intercellular transport of different kynurenine pathway metabolites. Given the obvious similarities in molecular structure between tryptophan and its metabolites, they might be transported via the same amino acid transporters. Indeed, LAT1 was also identified to transport kynurenine [65]. Furthermore, ASCT2 is co-expressed with QPRT (Figure 3), particularly in highly proliferative cells [66], which is in agreement with the highly proliferative phenotype of cytotrophoblasts and extravillous trophoblasts. ASCT2 might therefore facilitate transport of kynurenine pathway metabolites including quinolinic acid from/to neighboring cells and/or the circulation.

## 4. Kynurenine Pathway Functions

During pregnancy, the placenta creates a unique tolerogenic environment that prevents fetal rejection, whereas treatment of pregnant mice with the IDO inhibitor 1-methyl-tryptophan from the start of pregnancy resulted in fetal loss [6]. Starting this treatment later in pregnancy, after establishment of the fetus and placenta (E6.5), resulted in a high blood pressure, pathological placental appearance, and proteinuria in one out of the five mice, with accompanied trends towards reduced fetal and placental weights [67]. A similar preeclampsia-like phenotype was observed in IDO1-knockout mice [68]. These mice showed fetal growth restriction in the offspring, as well as renal dysfunction and impaired endothelium-dependent vasodilation in the mother, together with the absence of the normal pregnancy-associated decrease in systolic blood pressure and increase in heart rate Altogether these studies prove the importance of IDO1 in pregnancy and fetal development. The following section describes the function of the kynurenine pathway in normal placental development and function.

### 4.1. Immune Regulation

Figure 3 reveals that kynurenine pathway enzymes are not present in most of the immune cells residing within the placenta, suggesting the kynurenine pathway has no intracellular regulatory functions in these cells. Only IDO1 is expressed in certain dendritic cells in the first trimester placenta (Figure 3A) and co-expressed with the macrophage marker CD68 in immunohistochemistry (Table 2). Feto-maternal tolerance involves tight regulation of both fetal trophoblasts and maternal immune cells as well, and the decidua contains many maternal immune cell subsets.

The essence of IDO1 in pregnancy and fetal development was proposed to be due to its ability to regulate T cells [6,11]. Hence, most investigations have focused on this specific aspect of IDO1 function. Initially, IDO1 was simply thought to locally deplete tryptophan and thereby inhibit immune responses [11]. More recently it was acknowledged that tryptophan depletion in the placenta would be highly disadvantageous for fetal development [69]. Instead, tryptophan is utilized to form kynurenine metabolites with immune regulatory functions. Several metabolites can activate the AHR, a positive feedback regulator of the kynurenine pathway. Activation of the AHR by kynurenine induced the expression of IDO1 and interleukin (IL)-10 on dendritic cells [70,71,72]. Dendritic cells are important interactors for T cells and B cells. Indeed, their AHR-induced activation led to formation of FoxP3+ regulatory T cells (Tregs) [70,71,72]. The kynurenine pathway metabolites 3-hydroxyanthranilic acid and quinolinic acid were also able to directly regulate T cell function. These compounds directed T cell differentiation towards Tregs instead of T helper (Th)1 and Th17 cells, and selectively induced Th1 but not Th2 apoptosis [73,74,75]. Nevertheless, the application of the IDO1 inhibitor 1-methyl-tryptophan at day E6.5 in mice did not affect regulatory T cell numbers, even though it did cause pregnancy complications [67]. Now it has become apparent that this inhibitor can have multiple off-target effects, including increased kynurenic acid formation [76], potentially through activation of the AHR [77], interference with TLR signaling [78], and inhibition of tryptophan uptake through competition for the SLC7A5 transporter [60]. These effects may all have contributed to the observed pregnancy complications in animal studies with this inhibitor. Remarkably, IDO1 not only has an enzymatic role, but it can also function as a signaling molecule, as recently discussed by Pallotta et al. [79]. In that role, IDO1 shifts from the cytosol to early endosomes and functions in transforming growth factor (TGF)-β induced noncanonical signaling to establish a long-term immunoregulatory phenotype in dendritic cells [80,81]. This signaling function of IDO1 might provide an additional mechanism to maintain a tolerogenic environment in the placenta and decidua.

### 4.2. Vasodilation

The placenta does not have any neuronal innervation, hence its vascular perfusion depends entirely on the presence of circulating and locally secreted vasoactive substances [82]. Endothelial IDO1 facilitates tryptophan-induced vasodilation in placental chorionic plate arteries [19,83]. However, the mechanisms underlying tryptophan-induced vasodilation remain unclear. Although initially kynurenine, produced by IDO1, was identified as the vasodilatory mediator [17], recently, the vasodilator effect of tryptophan was attributed to IDO1- and H_2_O_2_-mediated cisWOOH formation [18]. However, we showed that in the placental chorionic plate arteries, tryptophan-induced vasodilation is not cisWOOH, but nitric oxide synthase dependent [19]. Worton et al. [84] found that direct application of kynurenine did induce vasodilation in human myometrial and omental arteries through activation of large-conductance Ca^2+^-activated K^+^ channels, but not in placental chorionic plate arteries. For all these studies, it is important to note that vasodilator effects could only be observed at very high concentrations: tryptophan >3 mM, kynurenine >0.5 mM. This would require a >100-fold increase of the in vivo plasma concentrations, approximated to be ~34 μM and ~61 μM for tryptophan, and ~1 μM and ~4 μM for kynurenine in the maternal vein and umbilical vein respectively, based on Table 1. Although these data could be interpreted to suggest that the kynurenine pathway does not have a physiological role in the regulation of placental perfusion, an alternative explanation is that the lack of effect is due to the limited uptake of tryptophan and kynurenine in in vitro models. Indeed, we showed that application of the lipid soluble ethyl ester of tryptophan resulted in increased vasodilation starting from a lower concentration [19]. In contrast, it was necessary to pre-incubate chorionic plate arteries with IFN-γ and tumor necrosis factor-α to observe vasodilation in response to tryptophan [19,83] Such IFN-γ pre-incubation was required to induce the expression of a tryptophan-selective transporter [63], demonstrating that not IDO1, but tryptophan uptake was a limiting factor for its vasodilator function. Future studies are needed to investigate the role of the transport system and its interaction with the kynurenine pathway, but as a first step it is essential to pinpoint the yet unidentified tryptophan transporter.

### 4.3. Placental Vascular Development

Placental development relies on new blood vessel formation to increase the feto-maternal surface area to allow sufficient oxygen and nutrient exchange. A role for IDO1 in vascular growth and development was first suggested in xenograft tumors in which neovascularization was enhanced by IDO1 overexpression. In a cancer model, Ido1^−/−^ mice were shown to have a substantially lower pulmonary vascular density compared with Ido1-competent mice at baseline, predominantly at the level of small- and medium-sized vessels [15,16]. The loss of Ido1 was accompanied by reduced IL-6 levels, but only in the presence of IFN-γ [15]. Interestingly the pulmonary vasculature, like the placental vasculature, is among the few that express relatively high levels of IDO1 already under physiological circumstances. Therefore, it is likely that endothelial cells express IDO1 to promote the formation of new placental vessels, and that a decreased IDO1 function might impair placental development.

### 4.4. Pro- and Anti-Oxidant Roles of the Kynurenine Pathway

An increased placental oxygen concentration can alter tryptophan metabolism. IDO1, TDO and HAAO expression increase with elevated placental oxygen concentration [31], and O_2_ functions as a co-substrate for IDO1 [85]. At increasing O_2_ concentrations, placental explants correspondingly displayed combined increases in tryptophan consumption and kynurenine excretion [86]. Since the co-substrate was initially thought to be superoxide anion (∙O_2_^−^), IDO1 was believed to scavenge reactive oxygen species (ROS) [87]. Rather than by IDO1, the anti-oxidant properties seem to be provided by kynurenine pathway metabolites, including 3-hydroxykynurenine, xanthurenic acid, 3-hydroxyanthranilic acid, and kynurenic acid [21,88,89,90]. The potential anti-oxidant roles of kynurenine pathway metabolites in the placenta have been reviewed by Xu et al. [20] As the xanthurenic acid and kynurenic acid concentrations were very low or even undetectable in placental tissue [19], ROS scavenging in the placenta seems to be mediated principally through 3-hydroxyanthranilic acid and 3-hydroxykynurenine, which have the highest radical scavenging properties of all kynurenine pathway metabolites [91]. Conversely, angiotensin II infusion is associated with increased kynurenine pathway activation and endothelial cell apoptosis through 3-hydroxykynurenine mediated superoxide formation [23]. Attenuation of these effects in IDO1-deficient mice [23], shows that a kynurenine pathway disbalance might result in accelerated apoptosis.

In summary, the roles of the kynurenine pathway in the placenta include regulating the immune system, thereby creating a tolerogenic environment inducing vasodilation, aiding in vascular development, and maintaining oxygen homeostasis. Additionally, recent research suggests that tryptophan to kynurenine metabolism and subsequent activation of the AHR can inhibit glycolysis, and shift the intracellular metabolism towards increased fatty acid oxidation [92]. In any of these roles, kynurenine pathway deficiencies in the placenta can contribute to pathological pregnancy development.

## 5. Pathological Pregnancies

The kynurenine pathway can have various roles during placental development as described in the previous sections. In this section, the relevance of the kynurenine pathway for healthy pregnancy development, underlined by associations between several placenta-related pregnancy complications and alterations in the kynurenine pathway, will be discussed.

### 5.1. Recurrent Miscarriage

A study by Munn et al. [6] suggested that the kynurenine pathway was pivotal for healthy pregnancy conception and to prevent fetal rejection. Interference with the kynurenine pathway with 1-methyl-tryptophan starting early in pregnancy led to loss of all allogeneic fetuses in mice. Several studies were subsequently designed to investigate IDO alterations in human cases of miscarriage. IDO expression was reduced in the endometrium, decidua and placental villi of women with recurrent miscarriage [93,94]. Whereas in normal pregnancy IDO expression was associated with an increased T cell and NK cell number in the endometrium, this relation was absent in cases of recurrent miscarriage [93]. Additionally, the proportion of IDO expressing decidual monocytes and dendritic cells was lower in spontaneous abortions compared to normal pregnancies [95]. Those cells, as well as the peripheral monocytes and dendritic cells from spontaneous abortions, did also not upregulate IDO as much in response to IFN-γ and a soluble fusion protein of CTLA-4 and immunoglobulin Fc as compared to normal pregnancies. Interactions between the endometrium, decidua and placenta are important for successful placental development, and these studies suggest an essential role for IDO for the prevention of fetal loss and maintenance of pregnancy.

### 5.2. Preterm Birth

Preterm birth is sometimes considered a less severe form of miscarriage, with a pregnancy duration long enough to provide a viable baby. Already in 1979, tryptophan concentrations were measured in the umbilical veins of term and preterm born infants [37]. The placentas of preterm pregnancies displayed an increased expression of IDO1 that was further augmented in cases of preterm premature rupture of membranes (PPROM) [96]. Other kynurenine pathway alterations included increased expression of HAAO and the transporter LAT1, but decreased expression of the enzyme KYNU, although the fold change in the latter was minimal [96]. IDO2 and TDO expression were unaltered [96].

Although Manuelpillai et al. [48] observed an even further increased expression of IDO, TDO and KYNU in preterm pregnancies with intrauterine infection compared to without [48], Karahoda et al. [96] showed no influence of intrauterine infection on the overall expression pattern kynurenine pathway enzymes. Interestingly, the latter authors did identify an association between intra-amniotic and maternal inflammatory markers, and placental expression of tryptophan pathway genes [96]. This suggests that there is a relation between certain inflammatory markers and placental kynurenine pathway function.

The addition of lipopolysaccharides (LPS) to human placental explants, as a model for intrauterine infection, resulted in an increased kynurenine secretion into the medium [48]. However, in vivo, the kynurenine concentration was decreased in the umbilical cord of babies born prematurely [48], showing that LPS-induced inflammation does not represent preterm birth very well. Another study detected a decrease in the free tryptophan concentration in the umbilical vein of preterm delivered infants, but since the total tryptophan concentration was maintained, this was potentially caused by alterations in fetal albumin and/or NEFA concentrations, although unfortunately these were not measured [37]. To further understand a potential role of alterations in the kynurenine pathway in preterm birth, it is important to identify in which specific cell types the expression of kynurenine pathway enzymes is altered.

### 5.3. Preeclampsia

Preeclampsia affects 2–8% of all pregnancies, and is a severe condition characterized by maternal de novo hypertension after 20 weeks of gestation, evidence of maternal organ damage (e.g., proteinuria, elevated liver enzymes, pulmonary or cerebral edema) and/or fetal growth restriction [97,98,99]. To date, the only cure is delivery of the placenta, and with it an often premature infant. Preeclampsia is believed to originate from deficient placentation and impaired placental development [98,99]. Interestingly, the expression of IDO1 was lower in the placentas of women with preeclampsia and correlated with disease severity [19,47,83,100]. The release of kynurenine from preeclamptic explants into the medium was concomitantly reduced, and the whole metabolic secretion profiles of preeclamptic explants that were maintained at 6% O_2_ were similar to those of healthy explants maintained at 1% O_2_ [101]. Despite lower placental IDO1 expression, we reported an enhanced tryptophan-induced vasodilation in preeclamptic compared to healthy chorionic plate arteries [19]. Therefore, it seems that the enhanced relaxant response to tryptophan in preeclamptic arteries is unrelated to IDO1. More likely, it reflects an increase in facilitated tryptophan transport. Indeed, when we applied the lipid soluble ethyl ester of tryptophan to circumvent transporters, we found a reduced vasodilatory response in preeclamptic chorionic plate arteries, representing the reduced IDO1 function. An increase in tryptophan transport was also evidenced by an elevated tryptophan concentration in preeclamptic placentas [19]. Since the other kynurenine metabolite concentrations were not altered [19], it suggests that the kynurenine pathway function was largely unaffected. It should be noted that these alterations only apply to early-onset preeclampsia, since, conversely, the tryptophan concentration was decreased in the placentas of women with late-onset preeclampsia [102]. Transport rather than IDO1 activity was also suggested to be rate-limiting for placental tryptophan metabolism by Kudo and Boyd [64]. As described earlier, increased maternal tryptophan concentrations in preeclamptic compared to healthy term women are unlikely to reflect placental alterations. This might explain why the concentrations of tryptophan and kynurenine pathway metabolites did not differ between preeclamptic and healthy pregnancies at term in the maternal circulation, nor in the umbilical artery and umbilical vein [29,103]. However, kynurenine metabolite concentrations could still function as a biomarker, as kynurenic acid was increased in the circulation of women who later developed preeclampsia [104].

To summarize, placentas of women with preeclampsia display reduced IDO1 expression, but this does not affect its vasodilator function and the fetal tryptophan supply. Therefore, it seems unlikely that interference with the kynurenine pathway would significantly improve placental function in preeclamptic women at term. As previously described, however, IDO may also aid in the formation of new blood vessels, and therefore a reduced IDO1 expression and activity at the beginning of pregnancy might contribute to impaired placentation and placental development, possibly progressing into preeclampsia. Therefore, the kynurenine pathway might provide interesting targets for future preventive strategies.

### 5.4. Fetal Growth Restriction

In the condition of fetal growth restriction (FGR), the fetus does not reach its potential biological growth. Like preeclampsia, FGR can originate from poor placental development, and both pregnancy complications often occur together. Diet-induced FGR in damns did not affect maternal to fetal tryptophan transfer, however, the conversion of maternally intravenously injected tryptophan into quinolinic acid was impaired in FGR fetuses [105]. As quinolinic acid is as a main source of de novo NAD^+^ formation, reduced quinolinic acid levels may contribute to impaired placental development and functioning. Whether this was a result of deficient placental or fetal metabolism could not be determined in the study design. However, IDO1 expression is reduced in FGR placentas [31,83], although one study reported a lower IDO1 expression in FGR and preeclampsia, but not FGR alone [47]. Such reduced IDO1 expression might be a consequence of a hypoxic placental environment that might result from deficient placental development. Lower oxygen concentrations suppressed IDO1, TDO and HAAO expression and activity in human placental explants [31]. Interestingly, although IDO1 was reduced in both FGR and preeclamptic placentas, preliminary data showed that tryptophan-induced vasodilation was absent in FGR placentas, in contrast to the improved vasodilation in preeclamptic placentas [83,105]. These data point towards a difference in transporters involved in preeclampsia versus FGR.

## 6. Pharmacological Interventions

Given the multifactorial role of the kynurenine pathway, it is not surprising that several treatments targeting this pathway have emerged from different medical fields, such as the currently implemented IDO1 inhibitors to treat cancer. The following section summarizes kynurenine pathway-based therapeutic opportunities to treat pregnancy complications. Additionally, it lists how currently investigated kynurenine pathway interventions might affect the placental function and pregnancy development.

Essential amino acids, including tryptophan, play pivotal roles in fetal development. The decreased kynurenine and free tryptophan umbilical vein concentrations in preterm birth [37,48], and the reduced placental tryptophan content in preeclampsia [19] might be improved by supplementation. Multiple studies have investigated the effects of amino acid supplementation during pregnancy of which the molecular mechanisms were reviewed by Hussain et al. [106]. Tryptophan supplementation by itself improved embryo survival in pseudorabies virus-induced pregnancy failure [107]. Apart from direct effects, tryptophan and kynurenine metabolites may affect fetal programming as well. Such effect is exemplified by the development of hypertension in the offspring of mothers that were exposed to experimental chronic kidney disease, which could be prevented by tryptophan supplementation of these pregnant rats, and was accompanied by restoration of the chronic kidney disease-induced alterations in the offspring’s gut microbiome [108]. In ruminants, tryptophan supplementation, when given in the form of N-acetyl-L-tryptophan (ACT) to bypass microorganism metabolism in the rumen and allow uptake in the small intestine, improved fetal growth [109,110]. In a recent study, Worton et al. [84] investigated the potential use of kynurenine as an anti-hypertensive agent for preeclampsia. As previously described, kynurenine induced vasodilation in healthy and preeclamptic human arteries, in agreement with earlier reported blood pressure lowering effects in animals, and the effects of its precursor, tryptophan [17,19,83,84].

Altogether, available data suggest that tryptophan supplementation can improve fetal growth. However, the in vitro proven effect of both tryptophan and kynurenine would require a >100-fold increase of the in vivo plasma concentrations. Tryptophan loading in healthy volunteers gave rise to increased circulating kynurenine metabolites and lipid peroxidation, suggesting oxidative stress [111]. Moreover, high tryptophan concentrations were reported to increase AHR signaling and thereby inhibit porcine placental cell proliferation [112]. Given the presence of the kynurenine pathway in several tissues, and the limited knowledge on the physiological functions of kynurenine and its metabolites in vivo, treatment of people, and specifically pregnant women, with excessive high tryptophan or kynurenine concentrations requires caution.

In that respect, it is important to note that tryptophan transport into the cell is an important rate-limiting factor of the kynurenine pathway [19,64]. Similarly, fetal nutrient supply is limited by placental transport and metabolism. Instead of tryptophan supplementation, increasing tryptophan or kynurenine transport may be more effective. To bypass the placental transport system, Tchirikov et al. [113] investigated the effect of intra-umbilical amino acid supplementation via a subcutaneously implanted port system in human fetuses with severe FGR. This intravascular treatment was successfully executed, prolonged pregnancy, and increased fetal weight. However, instead of correcting, the fetal amino acid concentrations deviated even more from normal values after treatment. The authors attributed this to large differences between the commercial amino acid solutions used for infusion and the fetal physiological concentrations, which probably enhanced the amino acid disbalance after treatment [113]. Another way to circumvent amino acid transport is the use of lipophilic analogues. The ethyl ester of tryptophan was shown to bypass defective gastrointestinal neutral amino acid transporters in a child with Hartnup disease [114]. Furthermore, we showed that tryptophan ethyl ester was more potent in inducing vasodilation compared with tryptophan [19]. When given to pregnant women, tryptophan ethyl ester might not only affect placental perfusion, but simultaneously improve fetal tryptophan delivery as well. Future studies are needed to carefully design the best mode of tryptophan supplementation and weigh the beneficial and dangerous effects of tryptophan, or kynurenine supplementation during pregnancy.

The most well-known pharmacological kynurenine pathway intervention is inhibition of IDO1. In oncology IDO1 inhibitors are generally used in combination therapy, complicating the interpretation of the individual effects of IDO1 inhibition. However, a phase II trial with the IDO1 inhibitor epacadostat (INCB024360, Figure 5) as standalone treatment reported that the treatment was relatively well tolerated in patients with myelodysplastic syndromes [115]. This suggests that interference with IDO in non-pregnant subjects is not harmful. However, epacadostat was also shown to inhibit neovascularization in a mouse model of pulmonary metastasis [15]. Angiogenesis and neovascularization are important processes in placental development, therefore IDO1 inhibitors might be harmful during pregnancy, and we suggest avoiding these drugs in pregnant women.

Several kynurenine metabolites have neuroactive functions and have been linked to neurodegenerative diseases [116], and therefore interventions targeting other kynurenine pathway enzymes have been developed in the neurological field. It is important to keep in mind that interventions with any of these metabolites might subsequently affect fetal brain development as well. *N*-acetylcysteine (NAC) can inhibit the formation of kynurenic acid through inhibition of KAT-2 [117]. Although this can improve cognitive function [118], low placental KAT-2 expression deems it unlikely for NAC to have placental effects. Furthermore, NAC also has broad anti-oxidant effects, making it difficult to establish whether potential effects of NAC are mediated through inhibition of KAT-2. Depressive symptoms are believed to arise from an increased formation of toxic kynurenine metabolites inside the brain. A study in mice showed that administration of *L*-leucine, a competitor for the LAT1 transporters, decreased kynurenine levels in the brain and reduced depression-like symptoms. Leucine is a natural competitor for the LAT1 transporter [119], and therefore competes with tryptophan transport (Figure 5). Its anti-depressant effects are currently being tested in a phase 2 clinical trial in individuals with major depressive disorders (NCT03079297). Kynurenic acid might antagonize neurotoxic metabolites in the brain and is metabolized from kynurenine, but kynurenine is also a substrate for KMO to form 3-hydroxy-kynurenine. Inhibition of KMO may thus increase the flux of kynurenine towards kynurenic acid instead of 3-hydroxy-kynurenine. Indeed, the KMO inhibitor JM6 (Figure 5), was reported to increase kynurenic acid levels, and thereby lower neuropathic pain intensity in rats and reduce neurological symptoms in a transgenic mouse model of Alzheimer’s disease [120,121]. The kynurenic acid analogue 7-chlorokynurenic acid, but also 4-chloro-3-hydroxyanthranilic acid (4-Cl-3-HAA) and 4-chlorokynurenine (4-Cl-Kyn or AV-101), a prodrug of 4-chloro-3-hydroxyanthranilic acid and a potent inhibitor of HAAO that suppresses quinolinic acid formation (Figure 5), can modify N-methyl-D-aspartate receptor signaling. Even though AV-101 altered 3-hydroxykynurenine and kynurenic acid concentrations without affecting kynurenine and quinolinic acid concentrations, it did not improve clinical outcomes [122,123]. It is unclear how these compounds affect circulating kynurenine metabolites, but it is important to keep in mind that suppression of IDO1 or any of the subsequent kynurenine pathway enzymes would result in a reduced de novo NAD^+^ synthesis likely affecting placental function, and possibly altering the fetal kynurenine metabolite homeostasis.

Most currently available treatments are focused on inhibition of the kynurenine pathway, which in cases of suspected preterm labor, might be interesting. However, in all pregnancy complications, it seems more relevant to explore opportunities to stimulate kynurenine pathway functioning. As outlined above, kynurenine pathway activation in an important source of de novo NAD^+^ formation, and may improve vascular development and function, and reduce oxidative stress. However, recently, an IFN-γ-induced increase in intracellular kynurenine formation was linked to a decrease in intracellular NAD(H) levels [92]. Although this seems counterintuitive, these experiments were performed in human coronary artery endothelial cells only, and therefore agree with the spatial segregation of the kynurenine pathway enzymes that are essential for de novo NAD^+^ formation. Nicotinic acid (niacin) supplementation, which may increase NAD^+^ formation through the salvage pathway, is used to treat vascular disease [124]. In rabbits, nicotinic acid was shown to inhibit vascular inflammation and endothelial dysfunction, independent of its effect on lipid metabolism [125]. These effects are similar to the effects of activation of the kynurenine pathway, therefore it seems relevant to find ways to improve NAD^+^ formation through the kynurenine pathway.

## 7. Preclinical Models

While investigating the kynurenine pathway it is important to consider significant differences between potential preclinical models. As earlier described, LPS-stimulated placental explants do not seem to represent preterm birth very well. It would be easiest to investigate the IDO1 and IDO2 specific functions using animal (knock-out) models as suggested by Kudo et al. [39]. However, the placenta is the most species-specific organ and potential interspecies differences should be considered. In mice, IDO activity is very high around conception but disappears at the end of pregnancy, opposite to the human situation [126]. In rat placentas, Ido1 was not detected, but placental expression of Ido2 increased towards term, resembling the human IDO1 enzyme [58]. The choriocarcinoma cancer cell lines which are regularly used in placental research also seem unsuitable to investigate the kynurenine pathway, because of dissimilar gene expression of kynurenine pathway associated genes compared to primary trophoblast cells [27]. Therefore, it seems most relevant to investigate tryptophan metabolism through the kynurenine pathway in human models, or if using one of the above models, to be aware of the significant differences with the human situation.

## 8. Conclusions

This review for the first time identified spatial segregation between kynurenine pathway elements in the placenta, evidenced by distinct kynurenine pathway enzyme expression profiles between different placental cell types as summarized in Figure 4. Consequently, complete metabolism of tryptophan through the kynurenine pathway requires many cell types, and intercellular transport of the kynurenine metabolites. The placental kynurenine pathway has important roles as a local signaling pathway, providing a source of de novo NAD^+^ synthesis, and regulating tryptophan and kynurenine metabolite supply to the growing fetus. Tryptophan and metabolite transport are limiting for kynurenine pathway functioning. Therefore, interference with tryptophan and kynurenine transport might more effectively improve placental kynurenine pathway function compared to interference with metabolites or enzymes. Recent studies suggest that the placental kynurenine pathway does not contribute to maternally circulating metabolite concentrations. Kynurenine pathway alterations in the placenta are thus unlikely to affect circulating metabolite concentrations in the mother. Alterations in circulating metabolites, such as an elevated kynurenine concentration in preeclampsia likely reflect maternal hepatic alterations instead. Future studies are necessary to decipher the roles of the kynurenine pathway in placental development and pregnancy progression, and specific areas of interest are: (1) to decipher the physiological roles of the kynurenine pathway in the placenta, (2) to identify the cell type specific alterations of the kynurenine pathway enzymes and transporters in pregnancy complications associated with placental dysfunction, (3) to determine what causes alterations in circulating metabolites if not mediated through the placenta, and (4) how these alterations influence fetal development and neonatal outcome. Essentially, it is important to identify the functional consequences of kynurenine pathway alterations and its effects on fetal homeostasis in pregnancy complications such as recurrent miscarriage, preterm birth, preeclampsia, and FGR. This knowledge may provide new treatment targets and will hopefully allow the safe implementation of kynurenine pathway interventions in pregnant women.

## Figures and Tables

**Figure 1 ijerph-18-11545-f001:**
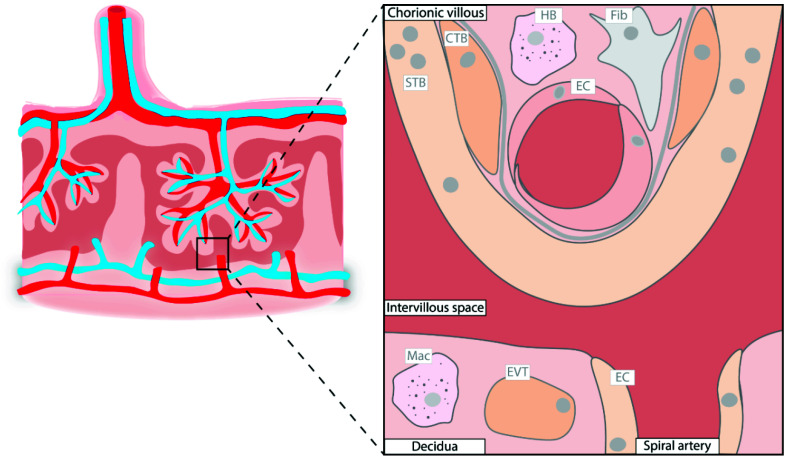
Schematic illustration of a term placenta with its most abundant cell types. CTB, cytotrophoblast; EC, endothelial cell; EVT, extravillous trophoblast; Fib, fibroblast; HB, Hofbauer cell; Mac, macrophage; STB, syncytiotrophoblast.

**Figure 2 ijerph-18-11545-f002:**
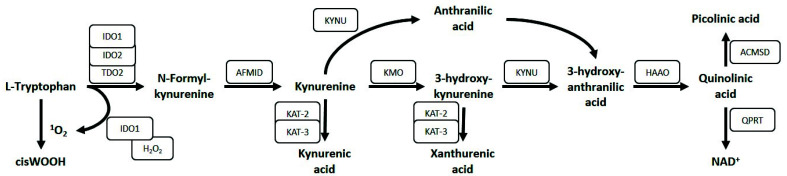
The kynurenine pathway. IDO, indoleamine 2,3-dioxygenase; TDO, tryptophan 2,3-dioxygense; AFMID, arylformamidase; KAT-2, aminoadipate aminotransferase; KAT-3, kynurenine aminotransferase 3; KYNU, kynureninase; KMO, kynurenine 3-monooxygenase; HAAO, 3-hydroxyanthranilate 3,4-dioxygenase; ACMSD, aminocarboxymuconate semialdehyde decarboxylase; QPRT, quinolinate phosphoribosyltransferase.

**Figure 3 ijerph-18-11545-f003:**
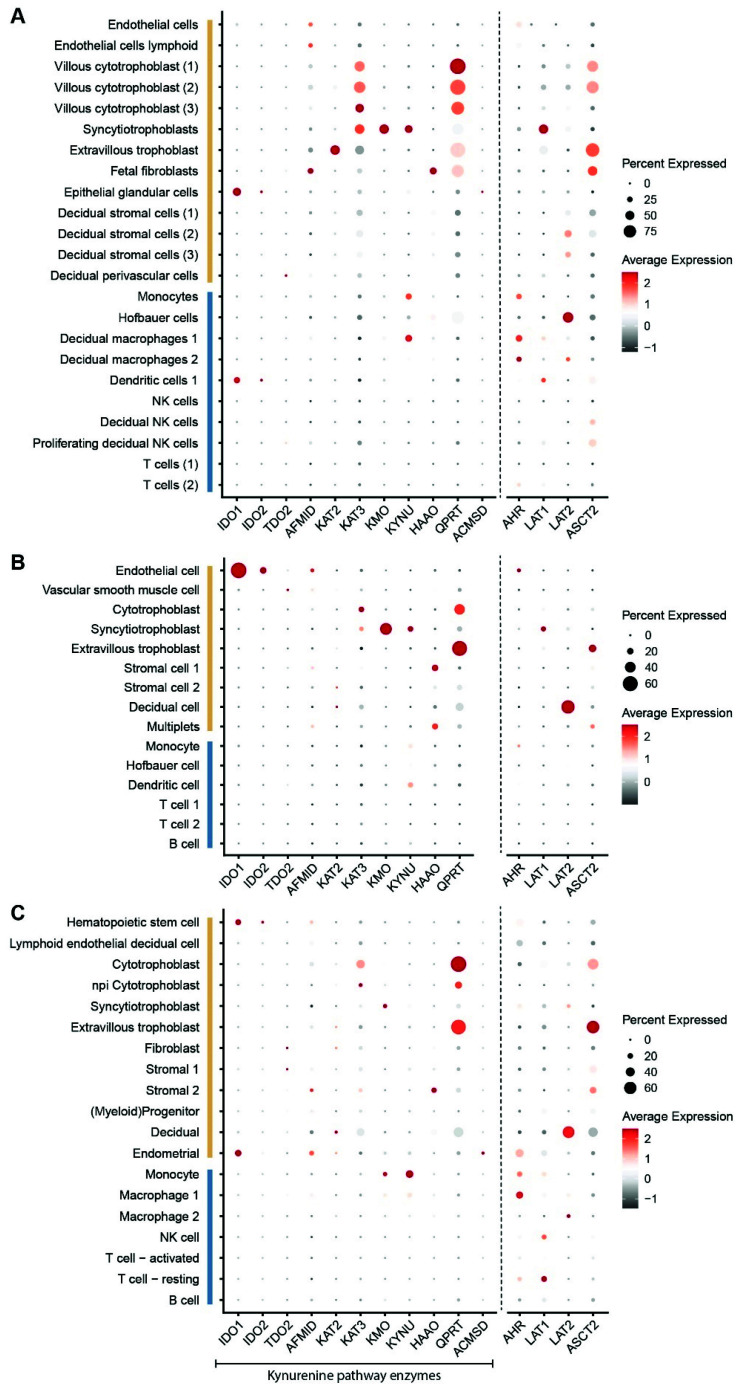
Cell type specific mRNA expression of kynurenine pathway enzymes, receptor, and transporters in placental tissue. Data were obtained using the single cell RNA sequencing data from first trimester placentas by Vento-Tormo et al. [49] (**A**), and term placentas by Pique-Regi et al. [50] (**B**) and Tsang et al. [51] (**C**). The cell types are clustered based on immune cells (blue) or non-immune cells (yellow). The size of the symbols reflects the number of cells within a certain cell type expressing the gene, whereas the color of the symbol reflects the relative expression of each gene. NK, natural killer; npi, non-proliferating interstitial; IDO, indoleamine 2,3-dioxygenase; TDO, tryptophan 2,3-dioxygense; AFMID, arylformamidase; KAT2, aminoadipate aminotransferase; KAT3, kynurenine aminotransferase 3; KMO, kynurenine 3-monooxygenase; KYNU, kynureninase; HAAO, 3-hydroxyanthranilate 3,4-dioxygenase; QPRT, quinolinate phosphoribosyltransferase; ACMSD, aminocarboxymuconate semialdehyde decarboxylase.

**Figure 4 ijerph-18-11545-f004:**
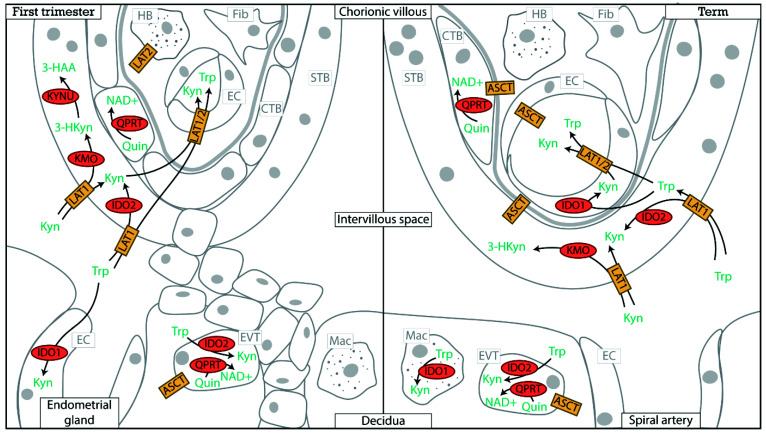
Illustration of kynurenine pathway enzymes and transporters localization in the first trimester and term placenta. This figure only shows the expression of enzymes (red) and transporters (yellow) confirmed by single cell RNA sequencing and/or immunohistochemistry data. The directions and interactions are a hypothetical display of the placental kynurenine pathway. How kynurenine pathway metabolites are exchanged between these cells, and whether this is facilitated by these, or other yet unidentified transporters is currently unknown. Enzymes (red): IDO, indoleamine 2,3-dioxygenase; KYNU, kynureninase; KMO, kynurenine 3-monooxygenase; QPRT, quinolinate phosphoribosyltransferase. Potential transporters (yellow): large neutral amino acid transporter (LAT); LAT1/2 depicts both LAT1 and LAT2 can be involved. Kynurenine pathway metabolites (green): Trp, *L*-tryptophan; Kyn, *L*-kynurenine; 3-HKyn, 3-hydroxykynurenine; 3-HAA, 3-hydroxyanthranilic acid; Quin, quinolinic acid; NAD^+^, nicotinamide adenine dinucleotide. Placental cell types in grey: CTB, cytotrophoblast; EC, endothelial cell; EVT, extravillous trophoblast; Fib, fibroblast; HB, Hofbauer cell; Mac, macrophage; STB, syncytiotrophoblast.

**Figure 5 ijerph-18-11545-f005:**
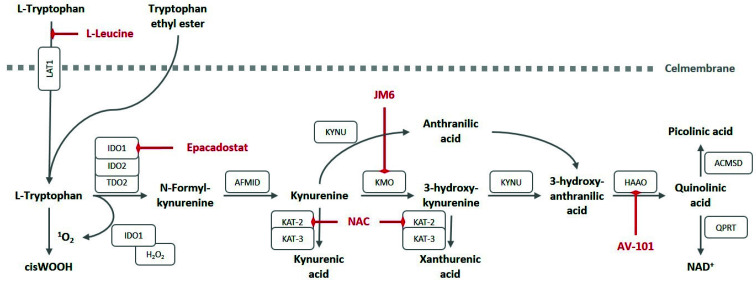
The kynurenine pathway with pharmacological interventions. L-Leucine is a natural competitor for Tryptophan transport by LAT1. The lipid soluble form of tryptophan, tryptophan ethyl ester, can circumvent the amino acid transporters. Epacadostat selectively inhibits IDO1. JM6 is an inhibitor of KMO. AV-101 (4-chlorokynurenine) is a prodrug of 4-chloro-3-hydroxyanthranilic acid, a potent inhibitor of HAAO. NAC (*N*-acetylcysteine) inhibits KAT2, and the subsequent formation of kynurenic acid and potentially xanthurenic acid. IDO, indoleamine 2,3-dioxygenase; TDO, tryptophan 2,3-dioxygense; AFMID, arylformamidase; KAT-2, aminoadipate aminotransferase; KAT-3, kynurenine aminotransferase 3; KYNU, kynureninase; KMO, kynurenine 3-monooxygenase; HAAO, 3-hydroxyanthranilate 3,4-dioxygenase; ACMSD, aminocarboxymuconate semialdehyde decarboxylase; QPRT, quinolinate phosphoribosyltransferase; LAT, large neutral amino acid transporter.

**Table 1 ijerph-18-11545-t001:** Tryptophan and kynurenine pathway metabolites concentrations in the maternal and fetal circulation at the end of pregnancy.

Measured Metabolite	Reference	Maternal Vein	Umbilical Vein	Umbilical Artery
Tryptophan (μmol/L)	Zhao (2021) [29]	34.5	64.1	∙
	Holm (2017) [30]	41.1	88.3	83
	Murthi (2017) [31]	28.5	60.2	58.9
	Schulpis (2008) [32]	28	26	∙
	Camelo (2004) [33]	29	40	∙
	Morita (1992) [34]	35.1	75.9	68.9
	Kamimura (1991) [35]	39.6	82.6	83.4
	Zanardo (1985) [36]	7.32	10.14	∙
	Tricklebank (1979) [37]	53.3	101.47	∙
Free tryptophan (μmol/L)	Kamimura (1991) [35]	7.5	17.5	16.4
	Tricklebank (1979) [37]	12.18	∙	20.46
N-Formyl-kynurenine (μmol/L)	Zhao (2021) [29]	0.18	0.40	∙
Kynurenine (μmol/L)	Zhao (2021) [29]	0.85	1.78	∙
	Murthi (2017) [31]	0.9	4	4
	Morita (1992) [34]	0.91	4.96	4.59
	Kamimura (1991) [35]	1.17	4.12	4.25
Kynurenic acid (nmol/L)	Zhao (2021) [29]	23	22	∙
	Kamimura (1991) [35]	20.5	38.1	39.8
	Murthi (2017) [31]	100	300	360
3-Hydroxykynurenine (nmol/L)	Zhao (2021) [29]	196	272	∙
Xanthurenic acid (nmol/L)	Zhao (2021) [29]	118	315	∙
	Kamimura (1991) [35]	188	295	258
Anthranilic acid (nmol/L)	Kamimura (1991) [35]	413	387	432
3-Hydroxyanthranilic acid (nmol/L)	Zhao (2021) [29]	287	448	∙
	Murthi (2017) [31]	300	300	340
	Morita (1992) [34]	N.D.	270	260
	Kamimura (1991) [35]	12	527	702
Quinolinic acid (nmol/L)	Zhao (2021) [29]	1333	2343	∙
	Murthi (2017) [31]	600	1500	1700
Picolinic acid (nmol/L)	Zhao (2021) [29]	223	287	∙
	Murthi (2017) [31]	500	1500	1400

N.D. not detected, ∙ not measured.

**Table 2 ijerph-18-11545-t002:** Summary of immunohistochemistry studies on placental indoleamine 2,3-dioxygenase (IDO) expression.

First Trimester	Kamimura [28] (1991)	Santoso [41] (2002)	Sedlmayr [42] (2002)	Hönig [46] (2004)	Kudoku [26] (2004)	Ligam [45] (2005)	Blaschitz [43] (2011)	Iwahashi [47] (2017)	Kudo [39] (2020)		Murthy [40] (2021)
Antibody against	IDO ^1^	IDO ^1^	IDO ^1^	IDO ^2^	IDO ^1^	IDO ^1^	IDO ^1^	IDO ^1^	IDO1 ^3^	IDO2 ^4^	IDO1 ^5^
*Placenta*											
Endothelial cells		nm	+	+			+	nm			
Macrophages		nm				+		nm			
Cytotrophoblasts		nm		+				nm			
Syncythiotrophoblasts	++	nm			++	++		nm		++	
Stromal cells		nm			+	+		nm			
*Decidua*											
Endothelial cells		nm	+	+			+	nm			
Epithelial glandular cells		nm	++	+	++	+	+	nm	++		
Extravillous trophoblasts		nm		++	++			nm			
Stromal cells		nm			+	+		nm			
Term											
*Placenta*											
Endothelial cells		++	+	+	++	++	++	++	++		++
Macrophages					++	++			+		
Cytotrophoblasts				+							
Syncythiotrophoblasts	++			+	++					+	
Stromal cells											
*Decidua*											
Endothelial cells				+			++		+		
Epithelial glandular cells				+							
Extravillous trophoblasts				++	+					+	
Stromal cells											

^1^ non-specific antibody (Dr. Takikawa, Kawasaki Medical School, Okayama Japan); ^2^ rabbit anti-IDO1 (Chemicon, Hofheim, Germany); ^3^ #86630 (Cell Signaling Technology, Danvers, MA, USA); ^4^ NBP2-46021 (Novus Biologicals, Centennial, CO, USA); ^5^ IDO1 rabbit mAb D5J4E (Cell Signaling Technology, cat. #86630); ++, strong staining intensity; +, moderate staining intensity; nm, not measured; empty cell, no expression detected.

## Data Availability

The following previously published datasets were used:

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
