# Peer review of "The Function of the Kynurenine Pathway in the Placenta: A Novel Pharmacotherapeutic Target?"

_ijerph, 2021, doi:10.3390/ijerph182111545_

Round 1

Reviewer 1 Report

The current article is an updated and expanded version of previous articles on the kynurenine pathway and placenta. The authors described the local kinetics of tryptophan, membrane transport mechanisms, its kynurenine metabolism, participation in the immune response, and the influence on the tension of the placental blood vessels. Very briefly, however, in the opinion of the reviewer, the subject of oxidative stress and kynurenine degradation products has been sufficiently discussed. As expected, the topic of pathological pregnancy and possible connections with kynurenines has been discussed in an expanded way. Chapter 6 (Pharmacological Interventions) is the least convincing part of the indicated review, only indirectly addressing the subject of pharmacological modulation of the placental kynurenine pathway. It seems that the chapter could be significantly reduced, especially with information on depressive states, neurological disorders, or cancer, and be more centered on the main topic. Unfortunately, the relatively destitute literature related to the pharmacology of the kynurenine pathway in perinatology and gynecology results in the fact that Chapter 6 does not provide significant information, nevertheless, the Authors should supplement their review with the latest articles published in 2020-2021 (over 60% of the cited works was published before 2015), therefore I would strongly suggest updating the literature.

Author Response

We thank the reviewer for these comments and suggestions. We have rewritten and shortened the section on Pharmacological Interventions from the neurological field. We feel that the remaining information on pharmacological interventions provides essential information  about the treatment options that are available for this pathway. Two articles about the kynurenine pathway in pregnancy were published just after our submission, and we have included those in this review now (Zhao et al. 2021 https://doi.org/10.1007/s43032-021-00759-0 , Guru Murthi et al. 2021 https://doi.org/10.1016/j.placenta.2021.09.008). These data have been added to Table 1 and Table 2 as well. Other recent articles we additionally included are: Pallotta et al. 2021 The FEBS Journal, and Lacono et al. 2020 EMBO reports, Blanco Ayala et al. 2021 Antioxidants (Basel), Yong-Hwa Lee et al. 2021 Circulation, and Keaton et al. 2019 Int J Tryptophan Research. To the best of our knowledge, we have now included all recent literature on pregnancy-related kynurenine pathway research.

Reviewer 2 Report

The authors of the review “The Function of the Kynurenine Pathway in the Placenta: A Novel Pharmacotherapeutic Target?” have described that IDO1 and IDO2 are differently expressed between placental cells and have outlined the effects of altered placental kynurenine pathway in different pregnancy complications. The correlation between tryptophan metabolizing enzymes, tryptophan metabolites and pregnancy are clearly depicted and well documented, in most cases, by recent literature.

Nevertheless, in the present paper it’s not considered the existence of the non-enzymic function of IDO1 that reprograms the expression profile of immune cells toward a highly immunoregulatory phenotype (clearly described in a recent state-of-the-art review). Thus I would recommend at least a brief discussion on the possible contribution of the signaling-mediated IDO1 function on pregnancy.

After all, the review is clear, linear and suitable for publication in the Int. J. Environ. Res. Public Health.

Author Response

We thank the reviewer for these comments and suggestions. We have now included a brief summary about the non-enzymatic function of IDO1 in section 4.1 Immune Regulation (lines 376-383)

Reviewer 3 Report

This is an excellent compilation of tryptophan catabolism through the kynurenine pathway on the placenta. The authors describe the levels of KP metabolites and enzyme expression on placentathat there are in the literaturealso explain the effect of these metabolites under pathological pregnancies. Howeverauthors should take in consideration the next points.

  1. In Figure 1 is mentioned KAT 2 and KAT 3, but never is explained the differences between these enzymesAlsoin the figure just appear KAT3
  2.  After each sentencethe point should be after the references.
  3. Figure 5 is not mentioned in the textand it does not explain it.
  4.  Authors should add a brief resume about the degradation of Trp throughout the kynurenine pathway, adding also how is it modulation. I suggest adding a table in which mention every enzyme and how it is modulated.
  5. If KP enzymes are expressed in placenta, add information about implication with the rest of KP enzymes not just IDO.
  6. Authors should add in detail the effect of KP metabolites in immunomodulation and tolerance, since there is vast information about that, but it is not clear here.
  7. As KP metabolites are neuroactive, the authors should add a brief implication on the developing brain.
  8.  It should be proposed the modulation of KYNA production in figure 5, which could be through the NAC.

Author Response

We thank the reviewer for these comments and suggestions. Please find our answers to your questions below.

  1. In Figure 1 is mentioned KAT 2 and KAT 3, but never is explained the differences between these enzymes. Also, in the figure just appear KAT3.

Thank you for this suggestion, we added a brief discussion on KAT-2 and KAT-3 at lines 236-240 and lines 264-265. As now written in the manuscript, we believe that the low levels of xanthurenic acid and kynurenic acid, suggest that these metabolites (and similarly the enzymes KAT-2 and KAT-3 required for their formation) do not have important roles in the placental kynurenine pathway. Therefore we do not elaborate on the potential functions of KAT-2 and KAT-3. However, following your suggestion, and to provide a complete overview of the kynurenine pathway we included both KAT-2 and KAT-3 in Figures 1 and 5.

2. After each sentence, the point should be after the references.

We thank the reviewer for noticing this, we have changed this throughout the whole document.

3. Figure 5 is not mentioned in the text, and it does not explain it.

We have added in-text references to Figure 5 at lines 670, 695, 702, and 709. Also we extended the legend of Figure 5 to explain the different inhibitors.

4. Authors should add a brief resume about the degradation of Trp throughout the kynurenine pathway, adding also how is it modulation. I suggest adding a table in which mention every enzyme and how it is modulated.

This is a very nice suggestion. However, the enzymes, substrates and cofactors have been nicely summarized in a review by Badawy in International Journal of Tryptophan research in 2017. Therefore we feel this would be redundant in the current review. We now refer to this review in the introduction (lines 112-113)

5. If KP enzymes are expressed in placenta, add information about implication with the rest of KP enzymes not just IDO.

We discuss all kynurenine pathway enzymes in section 2. Placental Expression of Kynurenine Pathway Enzymes (‘Placental IDO1 Expression’ in the previous version). In section 6. Pharmacological Interventions we summarize how interventions with separate items of the kynurenine pathway might differently affect its function, and how this translates to clinical implications.

6. Authors should add in detail the effect of KP metabolites in immunomodulation and tolerance, since there is vast information about that, but it is not clear here.

Our review includes the section 4.1 Immune regulation that discusses the role of the kynurenine pathway in immunomodulation and tolerance in the placenta. We added a little paragraph about how IDO1 might also function as signaling molecule apart from its role as enzyme (lines 376-383)

7. As KP metabolites are neuroactive, the authors should add a brief implication on the developing brain.

The reviewer raises a valid point, kynurenine metabolites might indeed affect fetal brain development. However, we believe that many of the kynurenine pathway metabolites can and will affect fetal development in multiple ways. Since in this review we focus on the kynurenine pathway in the placenta, we believe that we cannot go into depth on this topic. However, we now address this point briefly at lines 679-684.

8. It should be proposed the modulation of KYNA production in figure 5, which could be through the NAC.

We thank the reviewer for this suggestion, we have included the function of NAC in lines 684-689 and in Figure 5

Round 2

Reviewer 3 Report

Authors took in consideration all the comments and the manuscript improve substantially